# Trio deep-sequencing does not reveal unexpected off-target and on-target mutations in Cas9-edited rhesus monkeys

Xin Luo[1,2,3,10], Yaoxi He [1,2,3,10], Chao Zhang[4,10], Xiechao He[1,3,10], Lanzhen Yan[3,5,10], Min Li[1,2], Ting Hu[1,2], Yan Hu[1,2], Jin Jiang[1,2], Xiaoyu Meng [1,2], Weizhi Ji[6], Xudong Zhao[3,5], Ping Zheng[1,7]*, Shuhua Xu [4,7,8,9]* & Bing Su[1,3,7]*

CRISPR-Cas9 is a widely-used genome editing tool, but its off-target effect and on-target complex mutations remain a concern, especially in view of future clinical applications. Non-human primates (NHPs) share close genetic and physiological similarities with humans, making them an ideal preclinical model for developing Cas9-based therapies. However, to our knowledge no comprehensive in vivo off-target and on-target assessment has been conducted in NHPs. Here, we perform whole genome trio sequencing of Cas9-treated rhesus monkeys. We only find a small number of de novo mutations that can be explained by expected spontaneous mutations, and no unexpected off-target mutations (OTMs) were detected. Furthermore, the long-read sequencing data does not detect large structural variants in the target region.

[1] State Key Laboratory of Genetic Resources and Evolution, Kunming Institute of Zoology, Chinese Academy of Sciences, 650223 Kunming, China. [2] Kunming College of Life Science, University of Chinese Academy of Sciences, 100101 Beijing, China. [3] Kunming Primate Research Center, Chinese Academy of Sciences, 650223 Kunming, China. [4] Chinese Academy of Sciences (CAS) Key Laboratory of Computational Biology, Max Planck Independent Research Group on Population Genomics, CAS-MPG Partner Institute for Computational Biology (PICB), Shanghai Institutes for Biological Sciences, CAS, 200031 Shanghai, China. [5] Key Laboratory of Animal Models and Human Disease Mechanisms of Chinese Academy of Sciences, Kunming Institute of Zoology, Chinese Academy of Sciences, 650203 Kunming, China. [6] Yunnan Key Laboratory of Primate Biomedicine Research, Institute of Primate Translational Medicine, Kunming University of Science and Technology, 650500 Kunming, China. [7] Center for Excellence in Animal Evolution and Genetics, Chinese Academy of Sciences, 650223 Kunming, China. [8] School of Life Science and Technology, ShanghaiTech University, 201210 Shanghai, China. [9] Collaborative Innovation Centre of Genetics and Development, 200438 Shanghai, China. [10] These authors contributed equally: Xin Luo, Yaoxi He, Chao Zhang, Xiechao He, Lanzhen Yan. *email: zhengp@mail.kiz.ac.cn; xushua@picb.ac.cn; sub@mail.kiz.ac.cn

CRISPR-Cas9 has been widely used to facilitate efficient genome editing in model and nonmodel animals[1]. It also serves as a promising tool for correcting deleterious mutations causing human genetic diseases. However, the assessment of off-target and on-target effects is insufficient, making it unsafe to implement in clinical therapeutic settings.

The specificity of CRISPR-Cas9 relies on the designed 20 bp guide RNA (sgRNA) and PAM[2,3]. In the genome, there are often many sgRNA-like sequences. Consequently, CRISPR-Cas9 may generate nonspecific editing, leading to OTMs. Previously, most of the studies on off-target were carried out in rodents or human cells. Rodents serve as important animal models in preclinical studies. However, rodents have failed to show features of human disorders in many aspects. For example, the clinical symptoms involving high-level cognitive functions cannot be reproduced faithfully in rodent models[4]. By contrast, nonhuman primates (NHPs) are genetically and physiologically similar with humans. Macaque monkeys have been used in biomedical research and are among the highest primates that can be genetically manipulated (without serious ethical concerns) to construct preclinical models for Cas9-based therapies[5–7]. Hence, exploring the off-target activity in Cas9-edited monkeys becomes crucial for future clinical applications.

The off-target effect has been investigated using whole-genome sequencing (WGS) of Cas9-edited cells or animals[8–12]. However, previous studies mainly focused on the potential off-target sites predicted by sgRNA binding, not on a genome-wide evaluation of de novo mutations (DNMs). Recently, Schaefer et al. found plenty unexpected mutations using WGS of Cas9-edited mice[13] though the claim was challenged by several groups[14–19], and the latest trio sequencing of Cas9-edited mice did not see unexpected off-target activity[14].

Besides the off-target effect, the on-target complex mutations induced by the CRISPR-Cas9 system may also be a concern. A recent study reported large fragment DNA mutations (LFDMs), including large deletions, insertions, and complex rearrangements at the targeted sites in CRISPR-Cas9-edited cell lines[20].

To evaluate the situation in monkeys, we perform trio WGS and long-read sequencing of the target regions of Cas9-edited rhesus monkeys (Macaca mulatta). We also analyze the published trio WGS data of Cas9-edited cynomolgus monkeys (Macaca fascicularis)[21]. We did not observe any unexpected OTMs in the Cas9-edited monkeys.

## Result

### MCPH1 gene knockout rhesus macaques using CRISPR-Cas9.
We designed two sgRNAs to target exon2 and exon4 of MCPH1-a human autosomal recessive primary microcephaly gene that plays a key role in primate brain development and evolution[22–24] (Supplementary Fig. 1). Firstly, using zygotic injection of Cas9 mRNA and two sgRNAs, we achieved a high knockout efficiency for MCPH1 at embryo level. Among the 15 rhesus monkey embryos tested, 13 of them were knockout positive (86.6%), including 3 (20%) knockout homozygotes (Supplementary Fig. 2 and 3). To generate MCPH1 knockout monkeys, we microinjected 30 zygotes, among which 24 zygotes developed normally and were transferred into 6 surrogate females, resulting in two pregnancies of twins and triplets, respectively. The surrogate female with twins had premature delivery at 138-days gestation, leading to a live male monkey and a dead female monkey. We performed C-section at 160-days gestation for the other surrogate female with triplets, and all three monkeys (one male and two females) were alive (Supplementary Fig. 1 and Supplementary Table 1). We first used PCR-clone sequencing to evaluate the Cas9-editing status, and the result showed that all offspring

monkeys were successfully modified by Cas9 except for rmO4$^{ko*}$ from the triplets. The dead female monkey (rmO2$^{ko}$) was a homozygous knockout (Supplementary Fig. 1). Interestingly, all Cas9-induced mutations were located in exon2 of MCPH1. No exon4 mutations were detected although sgRNA2 was designed to target exon4 and showed a high efficiency in the embryo test (Supplementary Fig. 3). We speculate this might be due to technical problems such as misoperation of microinjection.

### Whole-genome deep sequencing and variant calling (VC).
The five Cas9-treated monkeys and their three wildtype parents were subject to WGS by the Illumina X10 platform ("Methods" section) (Fig. 1a and Supplementary Table 1). Blood samples were taken from the four live monkeys for DNA extraction, and for the dead monkey, multiple tissues (brain, liver, and muscle) were sampled. We achieved a median 46× depth of genome coverage (Table 1). The WGS data exhibited a high reads quality with Q30 > 85%, mean duplicate percentage of 12.03%, and properly paired reads >96% (Supplementary Table 2).

Using WGS data, we first reassessed the Cas9-editing efficiency at the MCPH1 locus (Table 1 and Supplementary Fig. 4). Consistent with the PCR-clone sequencing, we observed mosaic patterns of MCPH1 knockout for all Cas9-edited monkeys. The knockout efficiency ranges from 12.2 to 95.3% (Table 1) in the four Cas9-edited monkeys and no sequence change was detected in the knockout-negative monkey (rmO4$^{ko*}$).

We then performed genome-wide VC using four different tools, including GATK[25], Platypus[26], Freebayes[27], and Samtools[28] (Fig. 1b). VC, quality control (QC), site filtering (SF), genotype filtering (GF), and universal mask (UM) were performed to obtain high-confident variants ("Methods" section) (Fig. 1b). The overlapped variants from the four different calling tools were taken as the high-confident variants (Supplementary Table 3).

### Evaluation of off-target mutations.
With the use of the Speed-Seq[8], we predicted the potential off-target locations for the two MCPH1 sgRNAs. Among the 4807 predicted off-target sites, no mutation was observed in the four Cas9-edited monkeys, including the monkey (rmO2$^{ko}$) with multiple tissue samples (Supplementary Table 4). We also used another off-target predictor (Cas-OFFinder)[29], and no mutation was observed in the four Cas9-edited monkeys when up to seven mismatches between the on-target and off-target site was allowed (Supplementary Table 4). Hence, no off-target effects were detected at the predicted sgRNA-binding sites.

### Detection of DNMs using trio WGS data.
To evaluate the genome-wide off-target effect, we explored the pattern of DNMs using the trio WGS data by TrioDeNovo software[30] (Fig. 1b). We first validated the genetic relationship between the Cas9-treated monkeys and their parents[31] (see "Methods" section). The IBD (identity by descent) results agreed well with the known kinship (Supplementary Fig. 5). Initially, we obtained on average 1,365 candidate DNMs for each Cas9-treated monkey by running TrioDeNovo and overlapping the DMNs called by different tools (Table 1). We then performed multiple filtrations (see "Methods" section) for the overlapped DNM sets, and obtained on average 34 high-confident DNMs for each monkey (32 substitutions and 2 indels) (Fig. 1c, d; Table 1; Supplementary Fig. 6, 7; Supplementary Data 1). We then performed Sanger sequencing in one Cas9-edited monkey (rmO1$^{ko}$), and 36 of the 39 identified high-confident DNMs (92.3%) were validated (Supplementary Fig. 8; Supplementary Data 1), suggesting that the employed DNM calling pipeline was reliable. These high-confident DNMs can be explained by the known spontaneous mutation rates of primates

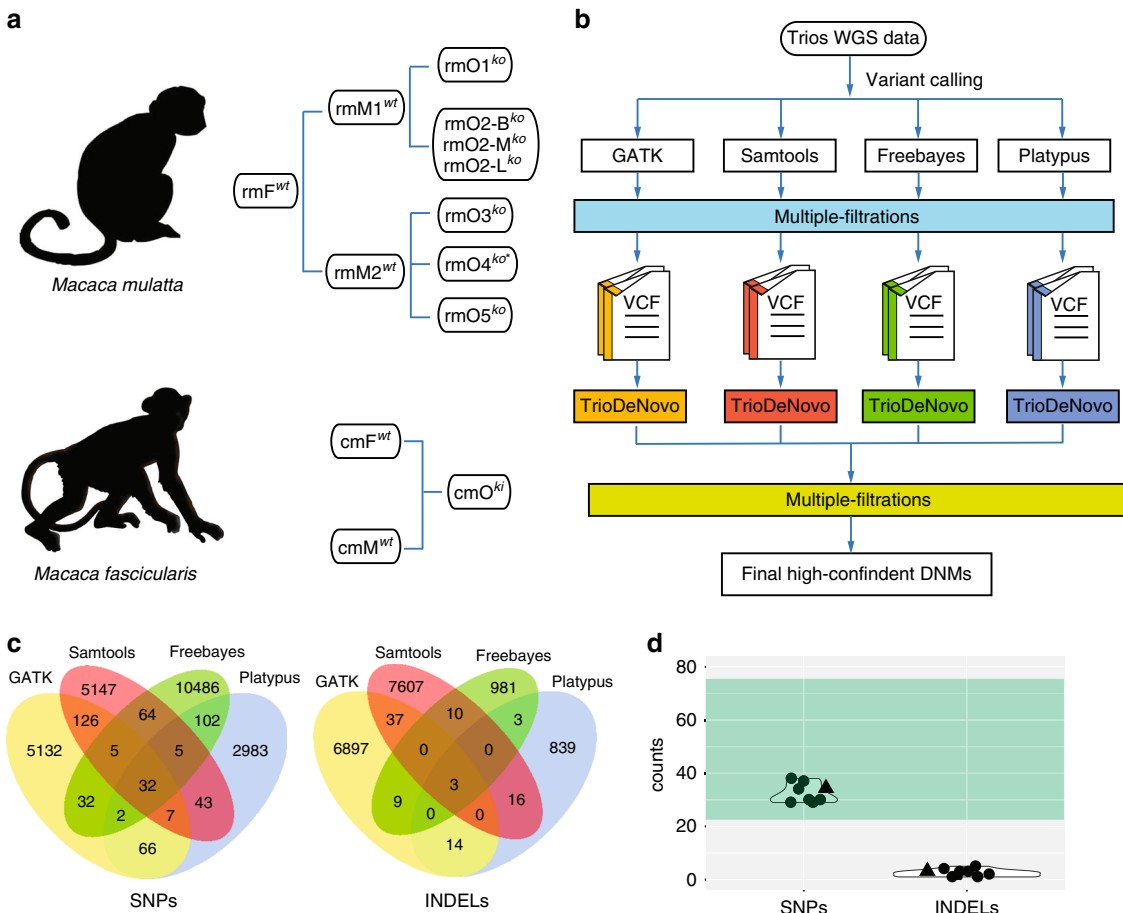

**Fig. 1** Detecting de novo mutations (DNMs) in the Cas9-treated monkeys. **a** The analyzed trios of rhesus monkeys (rm) and cynomolgus monkey (cm). F father, M mother, O offspring, ko knockout, ki knockin, B brain, L liver, and M muscle. **b** The pipeline of variant calling, filtering, and DNMs identification. HC.DNM-high confident DNMs. **c** Venn diagram of the high-confident DNMs for rmO2-B$^{ko}$ identified by overlapping candidate DNMs from four different calling tools. **d** Summary of the identified high-confident DNMs. The circles refer to Ca9-treated rhesus monkeys and the triangle refers to the Ca9-treated cynomolgus monkey. The green shadow indicates the range of expected DNMs (22–78) per generation given the known spontaneous mutation rates of primates.

**Table 1 Summary of trio WGS and identification of DNMs.**

| Monkey ID | Relationship | Treatment group | Depth | On-target ratio | Candidate DNMs | Final SNVs | Final INDELs |
|---|---|---|---|---|---|---|---|
| rmF$^{wt}$ | Father | Wildtype | 45.81 | – | | – | – |
| rmM1$^{wt}$ | Mother-1 | Wildtype | 46.23 | – | | – | – |
| rmM2$^{wt}$ | Mother-2 | Wildtype | 45.23 | – | | – | – |
| rmO1$^{ko}$ | Offspring-1 | Cas9-treat | 48.39 | 0.435 | 1324 | 34 | 5 |
| rmO2-B$^{ko}$ | Offspring-2-brain | Cas9-treat | 45.98 | 0.953 | 1169 | 29 | 3 |
| rmO2-M$^{ko}$ | Offspring-2-muscle | Cas9-treat | 44.85 | 0.938 | 1188 | 30 | 3 |
| rmO2-L$^{ko}$ | Offspring-2-liver | Cas9-treat | 45.35 | 0.952 | 1,230 | 29 | 2 |
| rmO3$^{ko}$ | Offspring-3 | Cas9-treat | 46.84 | 0.122 | 1267 | 38 | 1 |
| rmO4$^{ko*}$ | Offspring-4 | Cas9-treat | 45.92 | 0 | 1531 | 30 | 1 |
| rmO5$^{ko}$ | Offspring-5 | Cas9-treat | 47.60 | 0.321 | 1847 | 37 | 1 |
| cmF$^{wt}$ | Father | Wildtype | 64.08 | – | | – | – |
| cmM$^{wt}$ | Mother | Wildtype | 76.11 | – | | – | – |
| cmO$^{ki}$ | Offspring | Cas9-treat | 68.42 | – | 6060 | 32 | 3 |

(0.98–2.17 × 10$^{-8}$ per nucleotide per generation) with 22–78 expected DNMs per generation[32,33]. Consistently, we saw no correlation between the number of high-confident DNMs and the Cas9-editing efficiency ($R^2 = -0.140$, $P = 0.765$, Pearson's correlation test). In other words, the Cas9-editing efficiency does not affect the occurrence of DNMs in the Cas9-edited monkeys.

In addition, none of the detected DNMs are located or near (a 40 bp region around[8]) the predicted genome-wide off-target sites, ruling out the possibility of DNMs by the Cas9 off-target effect. To evaluate the statistical power of detecting DNMs, we adopted the previous method[14]. Given a median sequencing depth of 46.22× (Table 1) and a minimum allele frequency of 10%, the

power to detect one DNM occurred in the single-cell or the two-cell stage of zygote is at least 99.5% (see "Methods" section for more details). We also evaluated if there were de novo structural variants (SVs) (≥50 bp) using Delly[34] and Pindel[35] with multiple filters (refer to "Methods" section), and we did not see any high-confident de novo SVs in the Cas9-treated monkeys. (Supplementary Data 2).

To further confirm the DNM pattern seen in rhesus monkeys, using the same pipeline, we analyzed the published trio WGS data of a gene-knockin model via CRISPR-Cas9 in cynomolgus monkeys[21]. The trio included an *Oct4-hrGFP* knockin cynomolgus monkey and his two wildtype parents (Fig. 1a, b). The results showed that only 35 DNMs (32 substitutions and 3 indels) were detected (Fig. 1d; Table 1), concordant with the pattern seen in rhesus monkeys. For the sgRNA-predicted off-target sites, only one mutation (a 2-bp deletion) was seen in the knockin cynomolgus monkey as reported in Cai et al.[21].

**Examination of LFDMs**. Furthermore, to detect if there are LFDMs induced by CRISPR-Cas9 at the target regions, we PCR-amplified two ~6.0 kb regions covering exon2 and exon4 of *MCPH1*, respectively (Fig. 2a). The PCR products were sequenced by the PacBio platform (see details in "Methods" section). The results showed that no unexpected LFDMs were detected at the target regions in the Cas9-edited monkeys (Fig. 2a and Supplementary Data 2). It should be noted that a ~300 bp insertion was detected around the target region (exon4) in two Cas9-edited monkeys, which is also carried by their wildtype mother (without Cas9 treatment), suggesting this insertion in the Cas9-edited monkeys were inherited from their mother (Fig. 2b and Supplementary Data 1). Consequently, no unexpected LFDMs were detected at the target regions in the Cas9-edited monkeys.

## Discussion

In this study, we used multiple tools to call variants and we took the overlap as the high-confident variant set. It is known that the performance of different tools varies when conducting genome-wide VCs. Hence, a combination of different calling tools is necessary to identify high-confident variants from WGS data. Notably, the four different tools exhibited high consistency (>80%) for VCs (Supplementary Table 3).

For DNM identification, we initially identified ~1365 candidate DNMs for each monkey. We found that >90% of them were the types disobeying the Mendel's law due to unexpected allele combinations in the offspring monkeys, but the alleles were in fact present in the parents. For example, the genotypes of the parents are AA and GG respectively, and we saw AA or GG in the offspring instead of the expected genotype of AG. These candidate DNMs are most likely noise, not true DNMs, which may be caused by the technical bias of next-generation sequencing. The same scenario was also seen in the reported mouse data[14]. We filtered out these candidate DNMs by the DNM filtration procedure ("allele filtering", "Methods" section).

In addition, gene knockout and knockin are formed by different repair procedures in the cell. We analyzed trio sequencing data from both knockout (rhesus monkeys) and knockin (cynomolgus monkeys) monkey models, and we did not observe unexpected mutations in either model, suggesting that the homologous repair template does not induce OTMs. Our WGS data indicate that neither the knockout nor the knockin monkeys possess unexpected mutations. However, it should be noted that although WGS data is powerful in detecting DNMs, considering the minimum allele frequency for VC of 10% in this study, we cannot fully exclude the possibility of low-frequency (<5%) DNMs.

Furthermore, we did not find any LFDMs in the Cas9-edited monkeys, contradicting the previous report using cell lines[20]. We speculate that this discrepancy may result from different DNA repair mechanisms between in vivo and in vitro systems. In addition, in the data by Kosicki et al., ES cells and other cell lines were used, which may not apply to in vivo system. Also, Kosicki et al. introduced Cas9 and gRNA constructs targeting intronic and exonic sites of PigA using PiggyBac transposon system, and the observed LFDMs were possibly induced by transposase itself, not by the Cas9 system. Finally, natural mutations of some genes (often seen in cell lines) can also induce LFDMs. For example, Yu et al. reported that Dna2 nuclease deficiency could lead to large and complex DNA insertions at chromosomal breaks[36].

Due to the limited monkey resource, we did not acquire non-Cas9-treated littermate controls. Instead, we used the knock-negative monkey (rmO4$^{KO*}$) as a "proxy", and no difference was detected between the knock-negative and the Cas9-edited monkeys in view of DNM frequencies. With this experimental design, though unlikely, we cannot formally rule out the possibility that the DNMs are induced by Cas9. In addition, MCPH1 is essential for mitotic and meiotic recombination DNA repair and for maintaining genomic stability[37]. Targeting this gene might result in some confounding effect. Previous study reported that only the homozygous *Mcph1*-del mice showed defect of DNA damage repair[38]. Most of our Cas9-edited monkeys are mosaics (heterozygous) with on-target ratios ranging from 12.2 to 95.3%. If MCPH1 affected DNA damage repair in the Cas9-treated monkeys, we would have detected difference of DNM frequencies among the monkeys having different mosaic ratios. Our results showed that the detected DNMs were similar among the Cas9-treated monkeys (29–38 DNMs, Table 1), ruling out the potential influence of MCPH1 as a DNA damage regulator.

In conclusion, based on our systemic evaluation of off-target and on-target effects in the Cas9-edited monkeys, we did not detect unexpected mutations (OTMs and LFDMs). Given the presented data was a single set of genomes editing experiments in monkeys, more tests are necessary to fully evaluate the safety issue of gene editing in primates.

## Methods

**Animals**. All animals were housed at the AAALAC (Association for Assessment and Accreditation of Laboratory Animal Care) accredited facility of Primate Research Center of Kunming Institute of Zoology. All animal protocols were approved in advance by the Institutional Animal Care and Use Committee of Kunming Institute of Zoology (Approval No: SYDW-2010002).

**SgRNA design and in vitro transcription**. Based on the rhesus monkey reference genome (Mmul_8.0.1), two sgRNAs were designed to target the *MCPH1* gene with sgRNA1 targeting exon2 and sgRNA2 targeting exon4. The sequences of the two sgRNAs are (PAM in bold): sgRNA1: CCTATGTTGAAGTGTGGTCATCC; sgRNA2: TTACACAGATGCAGGACAGCTGG. The sgRNAs were cloned into PUC57-sgRNA vector (Addgene No. 51132) (Supplementary Table 5). The sgRNAs were transcribed by the MEGAshortscript Kit (Ambion, AM1354) after the vectors were linearized by DraI (NEB, R0129S). SgRNAs were purified by the MEGAclear Kit (Ambion, AM1908). Cas9 mRNAs were transcribed by the T7 Ultra Kit (Ambion, AM1345) after the pST1374-Cas9-NNLS-flag-linker vector (Addgene No. 44758) was linearized with AgeI (NEB, R0552S). Cas9 mRNAs were purified by the RNeasy Mini Kit (Qiagen, 74104).

**Zygote injection and embryo transfer**. Briefly, healthy female monkeys with regular menstrual cycles were used as oocyte donors for superovulation by intramuscular injection with rhFSH (Recombinant Human Follitropin Alfa, GONAL-F®, Merck Serono) for continuous 8 days, then rhCG (Recombinant Human Chorionic Gonadotropin Alfa, OVIDREL®, Merck Serono) on day 9. The oocytes were collected by laparoscopic follicular aspiration 36 h after rhCG treatment. The MII (first polar body present) oocytes were selected for in vitro fertilization and the fertilization was confirmed by the presence of two pronuclei. Fertilized eggs were injected with a mixture of Cas9 mRNA (20 ng/μl), sgRNA1 (10 ng/μl), and sgRNA2 (10 ng/μl) into cytoplasm using a Nikon microinjection system. The injected zygotes were cultured in the chemically defined, protein-free hamster embryo culture medium-9 (HECM-9) containing 10% fetal calf serum (Hyclone

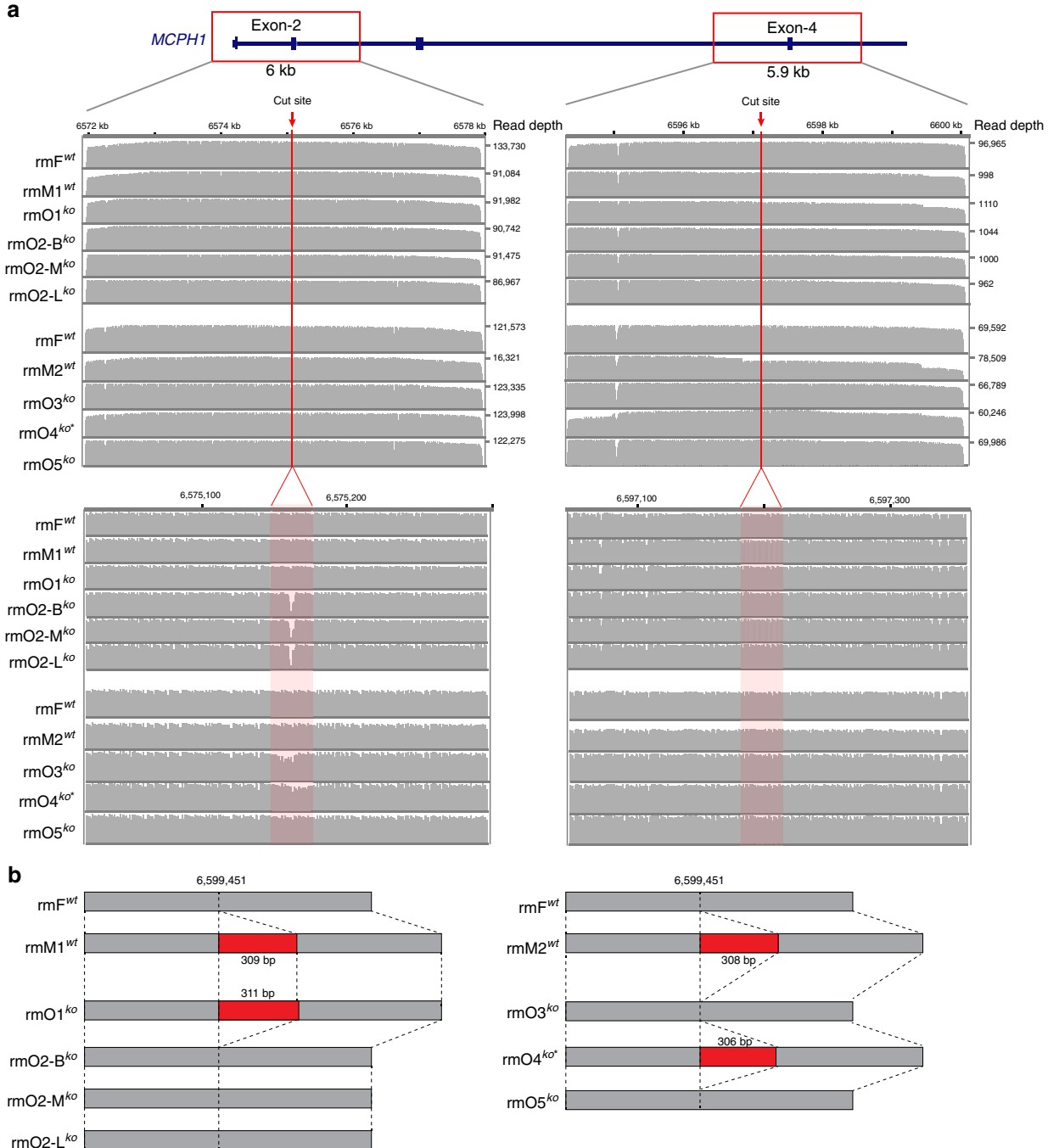

**Fig. 2** Long-read sequencing analysis of the sgRNA targeted regions. **a** The two sgRNA target regions of *MCPH1* (exon2 and exon4, respectively) tested by long-range PCR and PacBio sequencing. The upper and bottom panels show the coverage of the PacBio reads on the target regions based on whole-region view and cut-site region view, respectively. The cut sites of the sgRNAs in exon2 and exon4 are indicated with red arrow, red line, and shadow frame in red. **b** A ~300 bp insertion at the near-target region (2.3 kb from the cut-site) in two Cas9-edited monkeys (rmO1*ko* and rmO4*ko\**), which inherited from their wildtype mother (rmM1*wt* and rmM2*wt*).

Laboratories, SH30088.02) at 37 °C in 5% $CO_2$. The normally developed embryos from 2-cell to 8-cell with high quality were transferred into the oviduct of the matched recipients. A total of six monkeys were used as surrogate recipients, and typically, four embryos were transferred for each recipient female. The earliest pregnancy was diagnosed by ultrasonography about 30 days after transfer. Both pregnancy and number of fetuses were confirmed by fetal cardiac activity and presence of a yolk sac as detected by ultrasonography.

**DNA extraction**. Genomic DNA was extracted using the DNeasy Blood & Tissue Kit (Qiagen, 69506) according to manufacturer's instructions. The tissue samples included muscle, liver, and brain from rmO1*ko* and blood from the other monkeys. DNA samples were quantified using a NanoDrop spectrophotometer.

**Genotyping of *MCPH1* gene knockout rhesus monkeys**. PCR primers were designed to amplify the sgRNA targeting region (Supplementary Table 6). Targeted fragments were amplified with Taq DNA polymerase from genomic DNA. PCR products were subcloned into pMD19 vector (Takara, 3271). The colonies were picked up randomly and sequenced by M13-F primer.

**Whole-genome sequencing (WGS)**. WGS libraries were prepared using standard protocols for the Illumina X10 platform. Briefly, 100 ng DNA was fragmented

using a Covaris LE220 (Covaris), size selected (300–550 bp), end-repaired, A-tailed, and adapter ligated. Libraries were sequenced using the Hiseq X10 platform (Illumina) as paired-end 150 base reads. We generated on average 133 Gb raw sequence data per monkey. We performed QC by FastQC and GATK, and the mean Q30 of read-pairs are higher than 88%. Each sample has a raw read depth >46×. After masking the duplicates by Picard, we calculated the effective read depth (eDP) of the entire genome and the average eDP > 40. Mean percentage of PCR duplicates was lower than 13%, and the average mapped rate > 99.1%. The properly paired reads are >96% (Supplementary Table 2).

**Alignment and postalignment processing**. We used BWA MEM algorithm[39] to perform alignment, where short reads of rhesus monkeys (*M. mulatta*) were mapped to their reference genome (genome-build Mmul_8.0.1, rheMac8). The short reads of cynomolgus monkeys (*M. fascicularis*) were mapped to their reference genome (genome-build Macaca_fascicularis_5.0). The detailed command lines can be found in Supplementary Table 7. After the initial alignment, we run Picard's MarkDuplicates to remove duplicates in both datasets.

**Variant calling**. We called single-nucleotide variants (SNVs), and small insertions/deletions (INDELs), from de-duplicated bam files with GATK HaplotypeCaller[25], Platypus[26], Freebayes[27], and Samtools[28]. The command lines can be found in Supplementary Table 7. For GATK, variants were called and a GVCF file was generated for each sample, and then joint calling were performed for GVCF files of each trio, separately. For Platypus, Freebayes, and Samtools, we directly called the variants for each trio, separately.

**Variant filtering**. The overview of the variant filtering strategies can be found in Fig. 1b. We used SF, GF, and UM to filter against variants with low quality in each VCF set called by different callers. SF strategy filters variants at the site level, which takes QD (variant confidence/quality by depth) (QD > 2.0), mapping quality (MQ > 30), allele bias (AB > 0.1, at least Pval < 0.05), and strand bias (at least Pval < 0.05) into consideration. GF filters variants at the genotype level, which takes depth (15 < DP < 100 for SNVs) and genotype quality (GQ > 30) of each genotype into consideration. Since the information varies in the VCF sets generated by different VC tools, the corresponding SF and GF filtering variables and parameters are different. The details were summarized in Supplementary Tables 8 and 9. The UM is a sample independent mask that identifies complex regions in the reference genome where VC can be challenging[40]. In our analysis, the UM included three components: (1) mappability mask; (2) low-complexity regions; and (3) repeat regions. The command lines that generated (1) and (2) are provided in Supplementary Table 10. We merged the three sets of regions. SNVs and INDELs in the UM regions were filtered out, of which the detailed commands can be found in Supplementary Tables 8 and 9. The number of variants after each of the filters were listed in Supplementary Data 2.

**Target region analysis**. We extracted the reads, which aligned to the sgRNA-binding regions as well as the regions 100 bp upstream and downstream by Samtools *tview*[28]. We investigated Cas9-target effect at these reads, and the number of reads with deletions near PAM were counted in calculating on-target rate (Supplementary Fig. 4).

**Prediction of off-target sites of the *MCPH1* sgRNAs**. SpeedSeq[8] and Cas-Offinder[29] were used to evaluate the potential off-target sites of sgRNA. For SpeedSeq, genomic sites with "NGG" or "NAG" PAM motifs and ungapped alignment with up to five mismatches with sgRNA1 or sgRNA2 were defined as potential off-target sites. For Cas-Offinder, genomic sites with "NGG" or "NAG" PAM motifs and ungapped alignment with up to seven mismatches with sgRNA1 or sgRNA2 were defined as potential off-target sites. All prediction results are listed in Supplementary Table 4.

**Kinship validation**. To reduce computational complexity and linkage disequilibrium effect, we only included the independent variants of whole-genome to create the IBS (identical by state) matrix by PLINK 1.07[31] with argument: *indep-pairwise 50 5 0.2*. A total of 866,199 independent variants were included in calculating IBD by PLINK 1.07.

**DNM calling**. We adopted TrioDeNovo software to identified DNMs, using default settings and ran by each parent/offspring trio. After obtaining the results from running TrioDeNovo (Raw_DNMs), we took the overlapped set as downstream DNMs (noStrict_DNMs) detected in VCF sets of different callers. Then, we performed multiple filtrations to remove the false positive DNMs: (1) allele filtering: the DNM candidates where at least one allele was absent from parental genotypes (AF_DNMs); (2) dbSNP filtering: the DNM candidates must be absent from public SNV database (dbSNPBuildID = 150) (dbSNPF_DNMs); (3) cross filtering: the DNM candidates shared between offspring were removed (CF_DNMs). To compare the consistency of different callers, we intersected the DNMs after allele filtering with four tools, and the venn plots were provided in Fig. 1c and

Supplementary Fig. 7. The number of each filtering step was presented in Supplementary Data 2.

**Power evaluation in detection of DNMs**. To resolve mosaic DNMs in the genome, we employed the method from mouse study[14]. Considering our median depth (46.22×) of WGS (Table 1), we set a minimum required de novo allele frequency as 10% (we must observe at least 4 mutant allele reads out of 40 to call a DNM)[14]. The probability of not calling a mutation seen in 3 or fewer reads in this case is 0.005. In other words, the power of detecting a mutation occurred in a two-cell embryo is 99.8%.

**De novo SV detection using WGS short-read data**. We conducted genome-wide SV calling using Delly[34] and Pindel[35], and called de novo SVs for each trio. SV filtering was performed with the following criteria: (1) SV quality was evaluated with "PASS"; (2) SVs with precise breakpoints; (3) SVs with >10 supported reads; (4) SV length between 50 bp and 2 Mb (>2 Mb-length SVs were manually checked and we did not see any); (5) overlap the SV sets from Delly and Pindel (an overlapped SV was defined when the overlapped length is more than 50% of the reciprocal similarity). De novo SV filtering included the following criteria: (1) the shared SVs between parents and offspring were excluded (parent-inherit); (2) the SVs included in the public SV database or reported as common SVs in populations[41] (known SV filtering); (3) the SVs shared among the offspring were removed (cross-filtering). (4) IGV tool was used to manually check the candidate SVs and filtered out SVs according to the alignment and coverage situations. The results of the SV numbers of each filtering step are presented in Supplementary Data 2.

**Library preparation and PacBio sequencing**. The *MCPH1* gene targeted regions were amplified using primers MCPH1-E2-5939bp-f: 5′- GGCGGGGGGGATAA CGGTGCCGAAAG-3′. MCPH1-E2-5939bp-r: 5′-GACAGGCATTAGGGAGGTC AAACAAGGCTCTTAGGGTA-3′ and MCPH1-E4-5713bp-f: 5′-GTTTTCAAGG TTCATCATGTTGTCATCTGTATT-3′MCPH1-E4-5713bp-r: 5′-ATTGTTTATG ATTAGTGAGACGAAGGATTTGC-3′. The PCR was performed using LA Taq DNA Polymerase (ClonTech). We prepared libraries following the PacBio protocol, every five PCR products (with distinct barcode) are pooled together, DNA damage repair, EXO III and VII digestion, Two AMPure PB bead washes, annealing, binding, and sequencing.

**Long-read data analysis**. For long-read PacBio data, we applied NGLMR software to map PacBio subreads to the rhesus macaque reference genome (version: rhe-Mac8), and then Sniffles was used to call SVs from the bam file and the variants with support by at least ten high-quality reads were included in the downstream analysis[42]. SV filtering was performed with the following criteria: (1) Sniffle calling evaluated with "PASS"; (2) SV supported reads >5% of whole-region read depth; (3) SVs have precise breakpoints. De novo SV filtering included the following criteria: (1) The same SVs between parents and offspring were filtered out (parent-inherit). An overlapped SV was defined as same with overlapping length reaching at least 50% of reciprocal similarity; (2) the SVs shared between offsprings were removed (cross-filtering). We used IGV tool to visualize the alignment and coverage of long reads by using bam file from NGLMR. The SV numbers of each filtering step are listed in Supplementary Data 2.

**Reporting summary**. Further information on research design is available in the Nature Research Reporting Summary linked to this article.

## Data availability
The sequencing data generated in this study was deposited in NCBI BioProject PRJNA588331. The source data underlying Figs. 1–2, Supplementary Figs. 1–8, and Supplementary Table 4 are provided as a Source Data file. All other data are available from the authors upon reasonable request.

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

## Acknowledgements

We thank Yan Guo for her technical assistance in this study. We thank Xingxu Huang for providing plasmids. This study was supported by grants from the National Natural Science Foundation of China (31621062, 31730088, 91731303, 31525014, 31771388, and 31711530221), the Strategic Priority Research Program (XDB13010000), the Key Research Program of Frontier Sciences (QYZDJ-SSW-SYS009) of the Chinese Academy of Sciences, the Program of Shanghai Academic Research Leader (16XD1404700), the National Key Research and Development Program (2016YFC0906403), and the Shanghai Municipal Science and Technology Major Project (2017SHZDZX01).

## Author contributions

X.L. and B.S. designed the study; X.L., M.L., J.J., T.H., Y.H. and X.M. performed experiments; X.H., L.Y., P.Z. and X.Z. assisted in reproductive technique, microinjection, and animal care; Y.H, C.Z., X.L. and S.X. performed data analysis; W.J. provided the WGS data of the knockin monkeys; X.L., Y.H. and B.S. wrote the paper.

## Competing interests

The authors declare no competing interests.
