## [Peer Review File · Nature Communications]

Reviewers' Comments:

Reviewer #1:

Remarks to the Author:

In this study, the authors analyze on-target and off-target mutations in Cas9 genome edited rhesus monkeys by trio whole genome-sequencing. Rhesus macaques were edited by injecting Cas9 mRNA and two sgRNAs into the cytoplasm of one-cell rhesus zygotes. The experiment is designed to address the question raised by Schaefer, Mahajan et al. about the possibility of widespread mutations induced by Cas9. While the topic is of broad interest, there are a number of major concerns about the experimental design, analysis, and conclusions drawn from this data.

Major:

1. What is the estimated lowest limit of detection for mutations in these experiments? Whole-genome sequencing is comprehensive but a low-sensitivity method for detecting off-target mutations when performed at typical sequencing depths of 30-50X. The answer to this fundamental question should inform the analysis and discussion.
2. What is the basis for concluding that CRISPR-Cas9 is relatively safe in primates? This is a very broad conclusion based on a single set of genome editing experiments in rhesus monkeys which are really designed to detect widespread Cas9-induced mutations as initially reported by Schaefer et al., a report that has subsequently been retracted. The sequencing in this experiment is not adequate to detect low-frequency mutations and it is therefore important not to over generalize the results of these experiments.
3. Is the known role of MHCP1 as a DNA damage response protein and in G2/M cell cycle arrest relevant? Targeting this gene might result in some confounding effect. Homozygous mhcp1 null mice are born at a reduced rate. There may be biological selection for cells that are not homozygous knockouts of MHCP1. Another gene without a known role in DNA repair should be additionally tested or the limitations of the study associated with targeting a DNA damage regulator should be frankly discussed.
4. Can the authors explain the lack of activity for sgRNA2, even though it was active in initial embryo tests? Is this a technical issue with the experiments (possible if all injections were performed with the same in vitro transcribed gRNA source)?
5. Are the de novo mutations detected in or near sequences that are homologous to the sgRNA? Bona fide Cas9-induced off-target mutations have so far been convincingly detected at sites with up to 6 mismatches relative to the intended on-target sites. An analysis of the sequence regions containing de novo mutations for sequences closely matching the intended target sites will be important and is more direct than the current approach employed by the authors. It would be helpful to see the best possible alignment of the gRNA sequences in these regions. The most important mutations to consider in this analysis are indel mutations which are typically induced by Cas9.
6. Were comparisons made with non-Cas9 treated control littermates? This would be the best control to compare the frequency of mutations observed between edited and unedited animals.

Minor:

1. What is the Cas9 used in this study (wild-type, high-fidelity, bacterial species of origin)?
2. It is not clear how the authors computationally predicted off-targets; this should be described in more detail.

3. The rationale for selecting MHCP1 as the target in the study is not clear and should be addressed.

Reviewer #2:

Remarks to the Author:

The manuscript is in general well written. In particular providing parameter details for commands used for genomics are very well received. On the negative side, there are some limitations

a. Novelty: I appreciate the modest title and the refrain from the authors in over interpreting their result (except the final paragraph of main manuscript which should be rephrased). Also the additional work required for primate models in comparison to previous efforts in mouse model is commendable. However the added advantage of the current study beyond prior work (citations 7,11,13) is not clear. As the authors state, primates do provide a better phenotypic representation of human diseases than mice. However It is not clear why a different in vivo performance of CRISPR-Cas9 would be expected.

The 2nd last paragraph of main manuscript on differences in repair mechanism in in vitro versus in vivo models helped.

b. Generalisability: More detail is needed why these two loci used in the manuscript are representative of the remaining genome. This would help showcase that these results are generalisable to future experiments.

c. Experimental design: The absence of any control samples is a concern. It would have been helpful to include untreated offsprings as controls to get a more balanced estimate of mutation rate in a direct apples-to-apples comparison. Studies in human samples aimed towards estimating the mutation rate have shown significant variability in results based on experimental design and analytics. The authors use a generally accepted analytics pipeline for identifying de novo mutations. This pipeline is intended to find high confidence de novo mutations and is conservative in its assignment of the de novo status tag.

d. Power of detection calculation: The authors should include a power of de novo mutation detection calculation similar to citation 7.

e. Long read sequencing: Inclusion of SMRT sequencing to ascertain SVs is well intentioned. I assume the ~6kb region choice is based on SMRT sequencing in citation 20. However some of the structural variants they (citation 20) found in followup Sanger sequencing exceeded this size. I could not find the average coverage information for PacBio sequencing. In the absence of it, usage of 10 alternate allele read cutoff seems overly conservative. With trio NGS data the authors should also evaluate SV calls (in addition to snps and indels). In particular those at the target and predicted off target sites need to be closely examined.

f. Mosaic mutations: Similarly mosaic mutations should be evaluated. In particular at the targeted and predicted off target sites this should be evaluated using deep sequencing. For the rest of the genome, the existing data can be re-evaluated using the raw prefiltered variant call set. This would help differentiate parental mosaic mutations from potential cas9 induced ones. Also detailed coverage statistics of these sites should be provided. Additionally the Sanger plots of the validated sites should be included.

g. Indel realignment: Although this is not necessary for GATK HaplotypeCaller but it will have have an impact on indel detection using samtools.

h. Efficacy of filters: Additional supplementary table listing the number of variants removed as a

result of each of the filters should be included.

i. Phenotype: The study has targeted gene of clinical relevance in humans. It would be helpful for the community to know any phenotype observed as a result.

j. Supplementary Fig 4: Please include similar results for the other NGS libraries for both targeted sites.

Reviewers' comments:

Reviewer #1 (Remarks to the Author):

In this study, the authors analyze on-target and off-target mutations in Cas9 genome edited rhesus monkeys by trio whole genome-sequencing. Rhesus macaques were edited by injecting Cas9 mRNA and two sgRNAs into the cytoplasm of one-cell rhesus zygotes. The experiment is designed to address the question raised by Schaefer, Mahajan et al. about the possibility of widespread mutations induced by Cas9. While the topic is of broad interest, there are a number of major concerns about the experimental design, analysis, and conclusions drawn from this data.

Major:

1. What is the estimated lowest limit of detection for mutations in these experiments? Whole-genome sequencing is comprehensive but a low-sensitivity method for detecting off-target mutations when performed at typical sequencing depths of 30-50X. The answer to this fundamental question should inform the analysis and discussion.

Authors' response: With the use of the previous method (Iyer, et al. 2018), we evaluated the statistical power of detecting de novo mutations (DNMs) in our WGS data. Given the on-target ratios are 12.2%-95.3% in our *KO* monkeys (Table 1), we assumed a minimum DNM frequency of 10% (this was also used in the mouse study; Iyer, et al. 2018), the power of correctly calling a DNM occurred in one-cell or two-cell stage zygote is very high (>99.8%) (Figure R1). We have added the result of power analysis in the revised manuscript, and the related method part was also updated. Additionally, in calling DNMs, we adopted four tools (GATK, samtools, platypus and freebayes) to ensure the accuracy of variant calling.

Figure R1. Power evaluation of detecting DNMs using WGS data

The probability distribution of correctly calling a heterozygous variant at different sequencing depths) (fold coverage). The quality of the base calls was assumed at Q30 (https://www.illumina.com/Documents/products/technotes/technote_snp_caller_sequencing.pdf)

2. What is the basis for concluding that CRISPR-Cas9 is relatively safe in primates? This is a very broad conclusion based on a single set of genomes editing experiments in rhesus monkeys which are really designed to detect widespread Cas9-induced mutations as initially reported by Schaefer et al., a report that has subsequently been retracted. The sequencing in this experiment is not adequate to detect low-frequency mutations and it is therefore important not to over generalize the results of these experiments.

Authors' response: We understand the reviewer's point. In the revised manuscript, we rephrased the conclusion. It reads as "In conclusion, based on our systemic evaluation of off-target and on-target effects in the Cas9-edited monkeys, we did not see unexpected OTMs and LFTMs. Given the presented data was a single set of genomes editing experiments in monkeys, more tests are necessary to fully evaluate the safety issue of gene editing for primates."

3. Is the known role of MHCP1 as a DNA damage response protein and in G2/M cell cycle arrest relevant? Targeting this gene might result in some confounding effect. Homozygous mhcp1 null mice are born at a reduced rate. There may be biological selection for cells that are not homozygous knockouts of MHCP1. Another gene without a known role in DNA repair should be additional tested or the limitations of the study associated with targeting a DNA damage regulator should be frankly discussed.

Authors' response: MCPH1 is essential for mitotic and meiotic recombination DNA repair and for maintaining genomic stability (Liang, et al. 2010). However, only the homozygous Mcph1-del mice showed defect of DNA damage repair that was responsible for the infertility phenotype (Gruber, et al. 2011).

Most of our Cas9-edited monkeys are mosaics (heterozygous) with on-target ratios ranging from 12.2% to 95.3%, and one monkey (rmO4^{KO*}) was negative. If MCPH1 affected DNA damage repair in the Cas9-treated monkeys, we would have detected difference of DNM frequencies among the monkeys having different mosaic ratios. However, the detected DNMs are similar among all he Cas9-treated monkeys (29-38 DNMs, Table 1), ruling out the potential influence of MCPH1 as a DNA damage regulator. In addition, we analyzed the published cynomolgus monkey data, which was a knockin model of the Oct4 gene (Cui, et al. 2018), from which we detected a similar number of DNMs, a further support to the results in rhesus monkeys.

4. Can the authors explain the lack of activity for sgRNA2, even though it was active in initial embryo tests? Is this a technical issue with the experiments (possible if all injections were performed with the same in vitro transcribed gRNA source)?

Authors' response: We understand the reviewer's point. In the initial embryo test, sgRNA1 and sgRNA2 were both active. The lack of activity for sgRNA2 in the transgenic monkeys is intriguing. We speculate this might be due to technical problems such as improper storage of sgRNA2 or misoperation of microinjection etc.

5. Are the de novo mutations detected in or near sequences that are homologous to the sgRNA? Bona fide Cas9-induced off-target mutations have so far been convincingly detected at sites with up to 6 mismatches relative to the intended on-target sites. An analysis of the sequence regions containing de novo mutations for sequences closely matching the intended target sites will be

important and is more direct than the current approach employed by the authors. It would be helpful to see the best possible alignment of the gRNA sequences in these regions. The mouse important mutations to consider in this analysis are indel mutations which are typically induced by Cas9.

Authors' response: Following the reviewer's suggestion, we analyzed the intersection between all the detected DNMs and the 4,328 predicted off-target sites, and we did not see any DNMs located within or near the 1kbp regions upstream or downstream of the predicted sites. We added this information in the revised text.

6. Were comparisons made with non-Cas9 treated control littermates? This would be the best control to compare the frequency of mutations observed between edited and unedited animals.

Authors' response: We understand the reviewer's point. For monkeys, it is hard to collect such data due to limited monkey resource, in particular female monkeys at reproductive ages. **Fortunately, among the 5 Cas9-treated monkeys, one monkey (rmO4^{KO*}) was negative for MCPH1 knockout.** This monkey can serve as "a littermate control" who showed no difference from the other Cas9-edited monkeys in view of DNM frequency. Also, we collected published data about spontaneous mutation rate in primates (22~78 expected DNMs per generation) (Kong, et al. 2012) (Besenbacher et al. 2018). The observed high-confident DNMs (29-38) of the Cas9-treated monkeys are all within the expected range of spontaneous mutation.

Minor:

1. What is the Cas9 used in this study (wild-type, high-fidelity, bacterial species of origin)?

Authors' response: We used the well-known SpCas9 system (from *Streptococcus pyogenes*) in monkey editing.

2. It is not clear how the authors computational predicted off-targets; this should be described in more detail.

Authors' response: We predicted off-targets following a reported approach using their published scripts (Iyer, et al. 2015). In brief, all possible sites with homology to the 23 bp sequence (sgRNA + PAM) were retrieved by a base-by-base scan of the entire monkey genome, allowing for ungapped alignments with up to 5 mismatches in the sgRNA target sequence. The scan is made suitably efficient by streaming the genome sequence, and using bitwise (rapid) operations to check for the number of mismatches between the (sgRNA + PAM) sequence and the specific segments of the genome. The output of this scan is in bed format. Off-target sites for a set of paired Cas9 sites are identified from the genomic coordinates of all off-target target sites computed for individual Cas9 sites. In the revised manuscript, we added more details.

3. The rationale for selecting MCPH1 as the target in the study is not clear and should be addressed.

Authors' response: *MCPH1* is a key player in neurogenesis. Dysfunction of *MCPH1* in mice induce primary microcephaly. However, rodents and primates have huge differences in view of brain structure and neuron types. For example, primates have expanded OSVZ (outer subventricular zone) which contained a mass of neural progenitor cells, but rodents do not have it (Hansen, et al.

2010). In this study, the MCPH1 gene was selected in generating knockout monkeys so that the role of MCPH1 in human neurogenesis can be analyzed, which will be beneficial to brain developmental diseases such as autism and schizophrenia.

Reviewer #2 (Remarks to the Author):

The manuscript is in general well written. In particular providing parameter details for commands used for genomics are very well received. On the negative side, there are some limitations

1. Novelty: I appreciate the modest title and the refrain from the authors in over interpreting their result (except the final paragraph of main manuscript which should be rephrased). Also the additional work required for primate models in comparison to previous efforts in mouse model is commendable. However, the added advantage of the current study beyond prior work (citations 7,11,13) is not clear. As the authors state, primates do provide a better phenotypic representation of human diseases than mice. However, it is not clear why a different in vivo performance of CRISPR-Cas9 would be expected. The 2nd last paragraph of main manuscript on differences in repair mechanism in in vitro versus in vivo models helped.

Authors' response: The capacity of DNA damage repair is different between primates and rodents during fetal development. Recent studies have found that the overall expression patterns of genes in the BER (base excision repair), DDR (DNA damage response and checkpoints), and HR (homologous recombination) groups were significantly different between primates and mouse, but similar between monkey and human. In particular, the key HR genes (EME1, RAD51, RAD54L, RECQL, SHFM1, UBA2, and XRCC2) showed distinct expression patterns between primates and mouse. These genes have increased expression post ZGA in monkey and human but equally expressed prior to and post ZGA in mouse (Wang, et al. 2017). These results suggest that a monkey model can truly mimic the human DNA repair mechanism in order to provide a reference for the safety and efficacy of clinical therapy using the CRISPR-Cas9 system in the future.

2. Generalizability: More detail is needed why these two loci used in the manuscript are representative of the remaining genome. This would help showcase that these results are generalizable to future experiments.

Authors' response: The two loci are important genes for brain development and human brain diseases, both of which showcased the safety of Cas9-editing. We understand that more tests are needed in order to have a generalized conclusion on the Cas9 safety issue. We therefore rephrased the conclusion as “In conclusion, based on our systemic evaluation of off-target and on-target effects in the Cas9-edited monkeys, we did not see unexpected OTMs and LFTMs. Given the presented data was a single set of genomes editing experiments in monkeys, more tests are necessary to fully evaluate the safety issue of gene editing for primates.”

3. Experimental design: The absence of any control samples is a concern. It would have been helpful to include untreated offspring as controls to get a more balanced estimate of mutation rate in a direct apples-to-apples comparison. Studies in human samples aimed towards estimating the

mutation rate have shown significant variability in results based on experimental design and analytics. The authors use a generally accepted analytics pipeline for identifying de novo mutations. This pipeline is intended to find high confidence de novo mutations and is conservative in its assignment of the de novo status tag.

Authors' response: We agree with the reviewer that a littermate control would be ideal. However, as we explained above, monkey gene editing experiments are extremely expensive and time consuming. **Fortunately, among the 5 Cas9-treated monkeys, one monkey (rmO4^{KO*}) was negative for MCPH1 knockout.** This monkey can serve as “a littermate control” who showed no difference from the other Cas9-edited monkeys in view of DNM frequency. Also, we collected published data about spontaneous mutation rate in primates (22~78 expected DNMs per generation) (Kong, et al. 2012) (Besenbacher et al. 2018). The observed high-confident DNMs (29-38) of the Cas9-treated monkeys are all within the expected range of spontaneous mutation.

4. Power of detection calculation: The authors should include a power of de novo mutation detection calculation similar to citation 7.

Authors' response: We evaluated our statistic power of de novo mutation detection according to the previous method (Iyer, et al. 2018). Given the median sequencing depth of 46.22X (Table 1) and a minimum required de novo allele frequency of 10% (Methods), the expected power to detect a DNM occurred in single-cell or two-cell stage of zygote is more than 99.8% (see Figure R1 in the author's response to Reviewer-1's first question).

5. Long read sequencing: Inclusion of SMRT sequencing to ascertain SVs is well intentioned. I assume the ~6kb region choice is based on SMRT sequencing in citation 20. However some of the structural variants they (citation 20) found in follow up Sanger sequencing exceeded this size. I could not find the average coverage information for PacBio sequencing. In the absence of it, usage of 10 alternate allele read cutoff seems overly conservative. With trio NGS data the authors should also evaluate SV calls (in addition to snps and indels). In particular those at the target and predicted off target sites need to be closely examined.

Authors' response: We chose the ~6kb region because our target PCR region of exon2 and exon4 is around 6kb. As suggested by the reviewer, we investigated longer SVs (>6kb), and we did not find any. The average sequencing depth of long reads in exon2 and exon4 are 198,498X and 58,745X, respectively. Given the super-deep PacBio sequencing, it is proper to use 10 reads as the cutoff to filter SVs (also, the 10 reads cutoff is also the suggestive cutoff by *Sniffle* for calling high-confident SVs) (Sedlazeck, et al. 2018).

For NGS validation, we used *Delly* (Rausch, et al. 2012) and *Pindel* (Ye, et al. 2009) to call the SVs in exon2 and exon4 regions. Consistent with the long-read result, no de novo SVs were detected except for the known on-target mutations.

6. Mosaic mutations: Similarly mosaic mutations should be evaluated. In particular, at the targeted and predicted off target sites this should be evaluated using deep sequencing. For the rest of the genome, the existing data can be re-evaluated using the raw prefiltered variant call set. This would help differentiate parental mosaic mutations from potential cas9 induced ones. Also detailed coverage statistics of these sites should be provided. Additionally, the Sanger plots of the validated

sites should be included.

Authors' response: As described above, we calculated the statistical power and it is quite high for the monkey data. Given the median sequencing depth of 46.22X (Table 1) and a minimum required de novo allele frequency of 10% (Methods), the expected power to detect a DNM occurred in single-cell or two-cell stage of zygote is more than 99.8%.

We re-evaluated the targeted and the predicted off target sites using the raw prefiltered variant call set, by investigating the allele depth of alignment reads to check the mosaic situations. To resolve mosaic DNMs in the rest of the genome, we employed the method from mouse study (Iyer, et al. 2018). We have added this information and the coverage statistics of each site in the revised manuscript (Supplementary Table 5). The Sanger plots of the validated sites were also added in the revised version (Supplementary Figure 8).

7. Indel realignment: Although this is not necessary for GATK HaplotypeCaller but it will have an impact on indel detection using samtools.

Authors' response: Thanks for the point. We used samtools for detecting indels due to the following considerations. First, GATK HaplotypeCaller and samtools leverage different category of information: the former is a haplotype-based method while the latter is an alignment-based method. Both information is helpful for detecting indels. Second, according to systematical comparison between diverse callers (Hasan, et al. 2015; Neuman, et al. 2013), we found that there were no major difference between GATK HaplotypeCaller and samtools in the sensitivity and positive predictive value (PPV) for indel, when the read depth is as high as 40X (the average depth in our sample is 46.2X). Third, though both variant callers have limitations, accepting indels detected by at least two calling methods significantly increase the PPV (as high as 0.991 for simulation data) without a major effect on sensitivity (Neuman, et al. 2013). Therefore, we adopted the cross-validating by different tools (including samtools) as the strategy to detect de novo mutations.

8. Efficacy of filters: Additional supplementary table listing the number of variants removed as a result of each of the filters should be included.

Authors' response: As suggested by the reviewer, we added this information in the revised version (Supplementary Table 12)

9. Phenotype: The study has targeted gene of clinical relevance in humans. It would be helpful for the community to know any phenotype observed as a result.

Authors' response: MCPH1 is a key gene for brain development (Gruber, et al. 2011; Ke, et al. 2016). Phenotyping the Cas9-edited monkeys takes a long time and is still on-going. Hopefully, the phenotype data will be available in the near future.

10. Supplementary Fig 4: Please include similar results for the other NGS libraries for both targeted sites.

Authors' response: The revised version included the corresponding results (Supplementary Figure 4)

References

- Besenbacher, S., et al. Direct estimation of mutations in great apes reveals significant recent human slowdown in the yearly mutation rate. *bioRxiv*. (2018).
- Cui, Y., et al. Generation of a precise Oct4-hrGFP knockin cynomolgus monkey model via CRISPR/Cas9-assisted homologous recombination. *Cell Res*. 28(3):383-386 (2018).
- Gruber, R., et al. MCPH1 regulates the neuroprogenitor division mode by coupling the centrosomal cycle with mitotic entry through the Chk1-Cdc25 pathway. *Nat Cell Biol*. 13(11):1325–1334 (2011).
- Hasan MS., Wu X, and Zhang L. Performance evaluation of indel calling tools using real short-read data. *Human Genomics*. 9:20 (2015).
- Hansen, DV., et al. Neurogenic radial glia in the outer subventricular zone of human neocortex. *Nature*. 464(7288):554-561 (2010).
- Iyer, V., et al. Off-target mutations are rare in Cas9-modified mice. *Nat Methods*. 12(6):479 (2015)
- Iyer, V., et al. No unexpected CRISPR-Cas9 off-target activity revealed by trio sequencing of gene-edited mice. *PLoS Genet*. 14(7):e1007503 (2018).
- Ke, Q., et al. TALEN-based generation of a cynomolgus monkey disease model for human microcephaly. *Cell Res*. 26(9):1048-1061 (2016).
- Kong, A., et al. Rate of de novo mutations and the importance of father's age to disease risk. *Nature*. 488(7412):471-5 (2012).
- Liang, Y., et al. BRIT1/MCPH1 is essential for mitotic and meiotic recombination DNA repair and maintaining genomic stability in mice. *PLoS Genet*. 6(1):e1000826 (2010).
- Neuman, JA., O. Isakov, and N. Shomron. Analysis of insertion-deletion from deep-sequencing data: software evaluation for optimal detection. *Briefings in Bioinformatics*. 14(1):46-55 (2013).
- Rausch, T., et al. DELLY: structural variant discovery by integrated paired-end and split-read analysis. *Bioinformatics*. 28(18):i333-i339 (2012).
- Sedlazeck, JF., et al. Accurate detection of complex structural variations using single molecule sequencing. *Nat Methods*. 15(6):461-468 (2018).
- Wang, X., et al. Transcriptome analyses of rhesus monkey preimplantation embryos reveal a reduced capacity for DNA double-strand break repair in primate oocytes and early embryos. *Genome Res*. 27(4):567-579 (2017).
- Ye, K., et al. Pindel: a pattern growth approach to detect break points of large deletions and medium sized insertions from paired-end short reads. *Bioinformatics*. 25(21):2865-71 (2009).

Reviewers' Comments:

Reviewer #1:

Remarks to the Author:

In this revised paper, the authors have adequately addressed some of the major issues with the initial manuscript. Importantly, the conclusions have been appropriately tempered in the new discussion. However, many of the responses in their rebuttal are not incorporated into the text itself. The issues remaining are summarized below:

1. The limits of detection of standard whole genome sequencing for a range of mutation frequencies should be addressed more directly in the text. For example, off-target mutations can occur at frequencies orders of magnitude lower than on-target mutations. With minimum allele frequency for variant calling of 10%, there is only a 14.2% chance of detecting an off-target mutation that occurs on average in 5% of alleles. Whole-genome sequencing experiments have high-power to detect high-frequency de novo mutations but cannot exclude the possibility that low-frequency mutations may occur. This should be clearly noted.

2. The responses to original Reviewer #1, Questions #3, #4, and #6 should be clarified in the text itself. This will help readers to understand the author's rationale for the experimental design.

These questions are listed again below:

a. Is the known role of MHCP1 as a DNA damage response protein and in G2/M cell cycle arrest relevant?

b. Can the authors explain the lack of activity for sgRNA2, even though it was active in initial embryo tests?

c. Were comparisons made with non-Cas9 treated control littermates?

3. Related to #2, if used as a control, the interpretation of the monkey negative for MCPH1 knockout should be carefully discussed. With this experimental design, though unlikely, it is impossible to formally exclude the possibility that the de novo mutations are induced by Cas9 and this should be mentioned.

4. The response to question #5 (Are the de novo mutations detected in or near sequences that are homologous to the sgRNA?) is insufficient, because it is not clear that in silico algorithms accurately and comprehensively predict Cas9 off-target activity, particularly if they are limited to low numbers of mismatches. Cas9 off-target activity has been detected with up to 6 or 7 mismatches between the on-target and off-target site. This could be evaluated using CasOffinder up to 6-7 mismatches or by directly aligning the sites with detected de novo mutations against the intended target site.

Reviewer #2:

Remarks to the Author:

The authors have addressed most of my concerns/comments. I agree with them that although an experimental design including untreated litter mates would have been ideal. In the absence of it, using the negative sample provides the best proxy. Given the time and monetary costs of such inclusion it is not fair to ask for a change in experimental design.

A few more details in addition to what the authors have already included would be helpful.

Point 5:

A) Delly and Pindel SV calls should be extended to the whole genome and any variant calls at predicted off target sites examined. These details should be included in the manuscript or supplementary material.

B) Please include the results for number of variants removed by each filter for SV calls from short

read data in A) as well long read data.

Regarding point 6:

A) Minor point: Please include the proportion of variants falling within off target sites in Supplementary table 12.

Reviewer #1 (Remarks to the Author):

In this revised paper, the authors have adequately addressed some of the major issues with the initial manuscript. Importantly, the conclusions have been appropriately tempered in the new discussion. However, many of the responses in their rebuttal are not incorporated into the text itself. The issues remaining are summarized below:

1. The limits of detection of standard whole genome sequencing for a range of mutation frequencies should be addressed more directly in the text. For example, off-target mutations can occur at frequencies orders of magnitude lower than on-target mutations. With minimum allele frequency for variant calling of 10%, there is only a 14.2% chance of detecting an off-target mutation that occurs on average in 5% of alleles. Whole-genome sequencing experiments have high-power to detect high-frequency de novo mutations but cannot exclude the possibility that low-frequency mutations may occur. This should be clearly noted.

Authors' response: According to the reviewer's suggestion, we added a brief discussion to indicate the limitation of detecting low-frequency de novo mutations (Page 9, Line 22; Page 10, Line 1-3).

2. The responses to original Reviewer #1, Questions #3, #4, and #6 should be clarified in the text itself. This will help readers to understand the author's rationale for the experimental design. These questions are listed again below:

- a. Is the known role of MHCP1 as a DNA damage response protein and in G2/M cell cycle arrest relevant?
- b. Can the authors explain the lack of activity for sgRNA2, even though it was active in initial embryo tests?
- c. Were comparisons made with non-Cas9 treated control littermates?

Authors' response: Following the reviewer's suggestion, we have added these content in the text of the revised manuscript (Page 10, Line 19-22 for Question (a); Page 5, Line 13-14 for Question (b); Page 10, Line 14-18 for Question (c)).

3. Related to #2, if used as a control, the interpretation of the monkey negative for MCPH1 knockout should be carefully discussed. With this experimental design, though unlikely, it is impossible to formally exclude the possibility that the de novo mutations are induced by Cas9 and this should be mentioned.

Authors' response: We added a brief discussion about the use of negative control. In the revised manuscript, we stated that the knockout-negative monkey was used as a proxy to the non-Cas9-treated littermate control, and we cannot completely rule out the possibility that the de novo mutations are induced by Cas9 (Page 10, Line 14-18).

4. The response to question #5 (Are the de novo mutations detected in or near sequences that are homologous to the sgRNA?) is insufficient, because it is not clear that in silico algorithms accurately and comprehensively predict Cas9 off-target activity, particularly if they are limited to low numbers of mismatches. Cas9 off-target

activity has been detected with up to 6 or 7 mismatches between the on-target and off-target site. This could be evaluated using CasOffinder up to 6-7 mismatches or by directly aligning the sites with detected de novo mutations against the intended target site.

Authors' response: To address the reviewer's concern, we re-evaluated the potential off-target locations for the two *MCPHI* sgRNAs using CasOffinder (Bae et al. 2014; *Bioinformatics*) with 0-7 mismatches between the on-target and off-target site (both 5'-NGG-3' and 5'-NAG-3' PAM patterns were evaluated). Totally, 31,794 potential off-target sites were predicted by CasOffinder. We extended 40 bp (refer to Iyer et al. 2015) of each predicted off-target site and overlapped with all the DNMs. Consistent with our previous SpeedSeq's result, no DNMs are located or near the predicted off-target sites. We added this result and cited the CasOffinder paper in the revised manuscript. The number of the predicted off-target sites using CasOffinder was added in the revised Supplementary Table 4 (Page 6, Line 15-18; Page 16, second paragraph)

Reviewer #2 (Remarks to the Author):

The authors have addressed most of my concerns/comments. I agree with them that although an experimental design including untreated litter mates would have been ideal. In the absence of it, using the negative sample provides the best proxy. Given the time and monetary costs of such inclusion it is not fair to ask for a change in experimental design.

A few more details in addition to what the authors have already included would be helpful.

Point 5:

A) Delly and Pindel SV calls should be extended to the whole genome and any variant calls at predicted off target sites examined. These details should be included in the manuscript or supplementary material.

Authors' response: According to the suggestion, we conducted genome-wide SV calling using Delly and Pindel and called de novo SVs for each trio. Considering the limitation of SV detection using NGS short-read data, we adopted multiple filtering steps to ensure obtaining the high-confident SVs (Methods). After filtering, no de novo SV was detected in the four cas9-treated monkeys. We added these results and the corresponding method description in the revised manuscript (Page 8, Line 1-3).

B) Please include the results for number of variants removed by each filter for SV calls from short read data in A) as well long read data.

Authors' response: All information of each filtering step for SVs (both SV sets called by short-read and long-read data) were added in the revised Supplementary Table 6.

Point 6:

A) Minor point: Please include the proportion of variants falling within off target

sites in Supplementary table 12.

Authors' response: We added the number and proportion of variants falling within the off-target sites in the revised version of Supplementary Table 6 (previous Supplementary table 12).

Reviewers' Comments:

Reviewer #1:

Remarks to the Author:

My earlier concerns have been addressed in the manuscript.

REVIEWERS' COMMENTS:

Reviewer #1 (Remarks to the Author):

My earlier concerns have been addressed in the manuscript.

Author response: The reviewer did not raise any further issues.